

# Impact of freeze-thaw cycles on soil structure and soil hydraulic properties

Frederic Leuther[1], Steffen Schlüter[1]

[1]Department of Soil System Science, Helmholtz Centre for Environmental Research – UFZ GmbH, Halle, 06120, Germany

*Correspondence to*: Frederic Leuther (frederic.leuther@ufz.de)

**Abstract.** The ploughing of soils in autumn drastically loosens the soil structure and at the same time reduces its stability against external stresses. A fragmentation of these artificially produced soil clods during winter time is often observed in areas with air temperatures fluctuating around the freezing point. Farmers benefit from the structural transformation by frost action in terms of better seedbed preparation and improved hydraulic connectivity. Previous studies have mainly focused on the

effects of freezing and thawing on soil structure stability rather than on the impact on pore structure. From the pore perspective, it is still unclear (i) under which conditions frost action has a measurable effect on soil structure, (ii) what the impact on soil hydraulic properties is, and (iii) how many freeze-thaw cycles (FTCs) are necessary to induce soil structure changes.

The aim of this study was to analyse the cumulative effects of multiple FTC on soil structure and soil hydraulic properties for two different textures and two different initial structures. A silt clay with a substantial amount of swelling clay minerals and a

silty loam with less swell/shrink dynamics were either kept intact in undisturbed soil cores taken from the topsoil from a grassland or repacked with soil clods taken from a ploughed field nearby. FTCs were simulated under controlled conditions and changes in pore structure ≥48 µm were regularly recorded using X-ray µCT. After 19 FTCs, the impact on hydraulic properties were measured and the resolution of structural characteristics were enhanced towards narrow macro-pores with subsamples scanned at 10 µm.

The impact of FTC on soil structure was dependent on the initial structure, soil texture, and the number of FTCs. Frost action induced a consolidation of repacked soil clods, resulting in a systematic reduction in pore sizes and macro-pore connectivity. In contrast, the macro-pore systems of the undisturbed soils were only slightly affected. Independent of the initial structure, a fragmentation of soil clods and macro-aggregates larger than 0.8 to 1.2 mm increased the connectivity of pores smaller than 0.5 to 0.8 mm. The fragmentation increased the unsaturated hydraulic conductivity of all treatments by a factor of 3 in a pF

range of 2.0 to 2.5, while water retention was only slightly affected for the silt clay soil. Already 2 to 5 FTCs enforced a well-connected meso-pore system in all treatments, but it was steadily improved by further FTCs. This steady improvement in structural quality in terms of meso-pore connectivity is put at risk by milder winters in mid-latitudes due to global warming.



# 1 Introduction

Soil structure is shaped by various biotic and abiotic drivers, such as bioturbation or wetting/drying, but also by ploughing or

compaction (Rabot et al., 2018). In the mid-latitudes, where winter months are dominated by fluctuating temperatures around an air temperature of 0°C, frost is an important pedogenic agent on structure development, consolidation, deformation, and particle transport (Van Vliet-Lanoë and Fox, 2018). When farmers plough their fields in late autumn, they create a rough soil surface of soil clods that are exposed to temperature and moisture change throughout the winter. Soil cultivation in spring can be facilitated if the soil clods are broken up into a fine and fragmented soil structure by exposure to frost (Edwards, 2013). For

compacted soils, Jabro et al. (2014) evaluated freezing and thawing associated with typical winter weather conditions as the most effective and economical way to alleviate soil compaction in a clay loam. Soil fragmentation and creation of new pores by frost can change hydraulic properties of soils (Qi et al., 2006;Chamberlain and Gow, 1979) and reduce the risk of soil and nutrient loss in spring snowmelt (Deelstra et al., 2009). The prospect of milder and dryer winters in the mid-latitudes due to climate change (Kjellström et al., 2018) could influence this seasonal structural transformation.

The regular periodic alternation of radiation leads to temperature fluctuations in the top soil, the amplitude of which depends on the heat capacity of the soil and its thermal conductivity. Both properties vary among different soil constituents, which affects the freezing process of soils. Water has a low thermal conductivity compared to quartz and clay minerals but a high specific heat capacity (Bolt and Miller 1958). The heat transfer in soil is also dependent on the connectivity of the porous medium, which for solid matter is limited to single contact points. A high water content increases the area between particles

for heat transfer and thus promotes the heat transport to depths. As water is moving in such a porous system, additionally heat is transferred by convection. The different thermal conductivities of the soil constituents can lead to an uneven spatial distribution of ice crystal formation. A well-known example is the upheaval of stones to the soil surface due to their comparably high thermal conductivity and the resulting formation of ice crystals on their underside (Van Vliet-Lanoe and Dupas, 1991). During frost action, the resulting forces that lead to the disintegration of large aggregates by pressure release are strongly

dependent on the soil water content (Henry, 2007;Kværnø and Øygarden, 2006;Oztas and Fayetorbay, 2003). In saturated and nearly saturated systems, the volumetric extension of freezing water in soil pores by about 10% can reach a surface pressure ranging up to 460 kPa (Dagesse, 2013). This potential can only be reached when the extension of water is blocked by soil particles and entrapped air, thus in unsaturated soil freezing water can expand into empty larger pores. At the microscale, the temperature at which water freezes is dependent on the pore diameter and the amount of solutes, which govern matric potential

and osmotic potential, respectively (Ashworth and Abeles, 1984;Ren and Vanapalli, 2019). Experiments with defined filter pores sizes have shown that in pores smaller than 100 nm the freezing point of water was reduced to -0.8 °C and further to -4°C for pores with 15 nm in diameter. When ice crystals are built surrounding water flows towards them and accumulates at the freezing front (Hansson et al., 2004;Loch, 1982;Torrance et al., 2008). While the desiccation process during freezing may initially increase the stability of soil aggregates, structural breakdown may be associated to the rewetting process during

thawing (Dagesse, 2013).



Differences in heat transfer, freezing temperature of soil solution in pores and the affinity of water to move to the border of ice formation can induce local heterogeneities of ice formation processes, wetting and drying, and thus cause soil structure changes by frost. Previous studies on the impact of FTC on microstructure focused mainly on measurements of soil structure stability and not on the impact on the pore network. Based on wet sieving and stability measurements, it was shown that

aggregate stability decreased with number of FTCs and with increasing water contents. Soils with clay contents between 30 and 60 % were more susceptible towards stability deterioration than loamy soils with less clay (Oztas and Fayetorbay, 2003;Dagesse, 2013). Lehrsch (1998) observed an increase in stability after 2 to 3 FTCs with little changes thereafter for field-moist aggregates from different loamy soils. Ma et al. (2019) found a continuous and significant reduction in aggregate size distribution for different soil conditions after 1 to 30 FTCs. Six et al. (2004) summarized that the described differences in ice

formation at the microscale lead to opposing processes of aggregate degradation by wetting and aggregate stabilization by drying. Therefore, aggregate stability measurements on bulk samples would result in an average of both processes. On a soil column scale, Starkloff et al. (2017) determined via X-ray µCT scanning a reduction in macro-porosity (>140 µm pore diameter), pore thickness and their specific surface area after 6 FTCs for a silty clay loam and especially for a loamy sand. For fine-textured soils, the creation of new vertical shrinkage cracks increased the water permeability although a reduction in total

larger voids was observed (Chamberlain and Gow, 1979). In his review about FTC experiments in soil science, Henry (2007) remarked that many experiments used artificially rapid rates of temperature change and unrealistic minima of -20°C, using a typical laboratory freezer. Rapid freezing could reduce the impact of FTCs by promoting the formation of relatively small ice crystals (Henry, 2007). In addition, most studies were based on a small number of FTCs and neglected the chance that mechanical stress can accumulate with multiple FTCs.

The aim of this study was to analyse how multiple FTCs change the structure of two differently textured soils and which structural features promote the decay of clods as observed by farmers. The development of soil structure during the experiment, in terms of macro-pore characteristics and pore distances in the solid phase, was analysed by X-ray µCT at a resolution of 48 µm. At the end of the experiment, the results were complemented by structural information ≥ 10 µm obtained by X-ray CT scans of subsamples. Furthermore, the determination of the hydraulic properties contributed indirect information on the

development of the pore system towards the meso-pores. The experimental setup followed the recommendations of Henry (2007), i.e. the cumulative effect of 19 FTCs was investigated, the temperature amplitude was adapted to field conditions and the freezing process was conducted from top to bottom. The hypotheses were that multiple freezing and thawing (i) alters soil structure and hydraulic properties by the decay of soil clods and the creation of new pores, (ii) the magnitude of structure changes varies with the amount of swelling clay minerals, (iii) a long-term developed soil structure is more stable against

mechanical stresses imposed by FTCs than an unconsolidated structure produced by ploughing.



## 2 Materials & Methods

### 2.1 Field site, sampling and sample preparation

Soils from two different study sites in Germany were used, which differ in texture, organic matter content, and clay mineralogy. The soil in Giessen was characterized as a stagno-fluvic Gleysol on loamy-sandy sediments over gley (FAO taxonomy) where

the soil texture was a silt clay (20 % sand, 40 % silt, 40 % clay) and organic matter content was 4.46 % (Jäger et al., 2003). The volume fraction of clay minerals was ordered as vermiculite > vermiculite-illite interstratifications = illite > chlorite = kaolinite with overall high swelling capacity (Diel et al., 2019). The soil texture of the haplic Chernozem at Bad Lauchstaedt was a silty loam (11 % sand, 68 % silt, 21 % clay) and soil organic matter content was 2.05 % (Altermann et al., 2005). The order of clay mineral fractions was illite > kaolinite > vermiculite (Dreibrodt et al., 2002), the swelling capacity was

comparably lower. Undisturbed samples were taken at both field sites below the grass cover at a depth of 5 cm with 250 cm³ aluminium sample cylinders. The mean bulk density of the undisturbed silt clay samples was 1.06 g cm$^{-3}$ (sd. ±0.04), that of the silty loam 1.46 g cm$^{-3}$ (sd. ±0.06). In Bad Lauchstaedt, soil clods were taken from the surface of a ploughed bare fallow next to the grassland. In Giessen, clods were taken in the vicinity of the undisturbed soil cores from a ploughed grassland. Sampling took place in winter 2019/2020 before the first frost. Temperature profiles in 5, 10 and 20 cm depth below the soil

surface from November to April 2017/2018 and 2018/2019 are provided in the supplementary information (S1 and S2). Empty 250 cm³ cylinders were repacked with soil clods at a bulk density of 0.70 g cm$^{-3}$ (sd. ±0.01) for the silt clay and 0.85 g cm$^{-3}$ (sd. ±0.06) for the silty loam to simulate the loose packing at the very surface of a ploughed field (Daraghmeh et al., 2009). The repacking of the soil clods of a specific size range reduced the structural heterogeneity often observed in a ploughed field with a much larger range of clod sizes. This reduction in heterogeneity was required to enable a statistical analysis of the

structural development. The samples were not compacted in order to maintain the stability of the soil clods taken from the ploughed fields.

The repacked and undisturbed 250 cm³ soil cores were placed in a sandbox (Eijkelkamp, Netherlands) and capillary-saturated at 0 hPa water pressure at the lower boundary. Then the matrix potential was adjusted to -10 hPa for 48 h to achieve rather uniform saturation within one treatment and to drain the largest macro-pores (> 140 µm, according to Young-Laplace-

equation). Samples were covered by a lid, sealed with Parafilm to prevent water loss by evaporation throughout the experiment, and weighed. Controls were kept in a refrigerator at 4 °C and samples for FTC were randomly placed on the bottom of an insolation box (Fig. 1 (a)). The space between samples was filled with bubble wrap to prevent freezing from the side. A layer of air-filled bubble wrap was also placed on top of the samples to slow down the freezing process. The set-up was covered with freezer packs that had an initial temperature of -25°C. The temperature inside a soil column was measured every 10 min

in the centre of a control adjusted to the same water content using a Pt 100 thermometer (Omega Engineering GmbH, Germany) and a DT80 data logger (Thermo Fisher Scientific Inc., USA). The 0°C point of the thermometer was calibrated with ice water before the start of the experiment. Every two to three days, when soil temperature was above 0.5°C, the freezing packs were replaced to start the next FTC (Fig. 1 (b)). For each study site, seven repacked samples (R) and five undisturbed samples (U)





were subjected to FTCs, while five repacked samples (R control) and three undisturbed samples (U control) were kept as a
control.

After 19 FTCs, eight to ten subsamples (16 mm in diameter, 16 mm in height) per treatment were taken from the top of two
250 cm³ samples. To do so, the soil was slowly pushed out of the 250 cm³ cylinders from the bottom and in doing so sharpened
aluminium cylinders with a fixed position were gently pushed into the soil from the top. Before scanning, the subsamples were
dried for 6 to 10 hours at room temperature to drain water retained in the visible macro-pores ≥ 10 µm for optimal image
contrast in the X-ray CT scans. The targeted water loss was calculated by the treatment specific water retention curves to
correspond a water potential of -300 hPa.

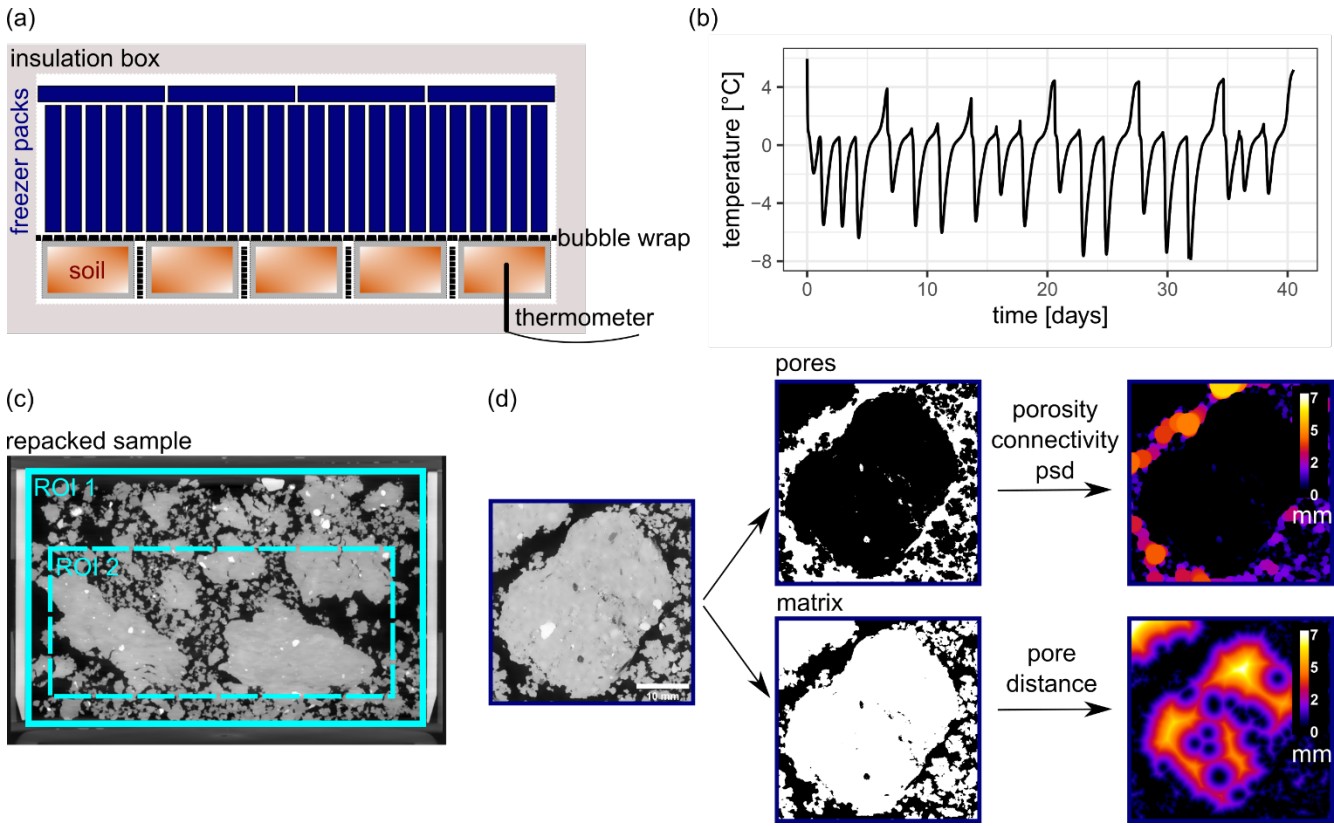

**Figure 1: Experimental setup of the study (a), the temperature profile during the experiment (b), vertical view of a repacked silt clay
sample with two different regions of interests for image analysis (c), and an overview of the different parameters determined via
image analyses to detect strutural transformation (d).**

## 2.2 X-ray CT and image processing

X-ray µCT scans were made after 0, 2, 5, 10 and 19 FTCs of both, control and FTC subjected samples, to track the cumulative
soil structure alteration. Imaging was done with an industrial X-ray µCT device (X-TEK XTH 225, Nikon Metrology,
Belgium). The 250 cm³ samples were scanned for 47 min using a 0.7 mm copper filter at 150 kV, 310 µA resulting in 2000
projections (0.708 s per projections). A voxel resolution of 48 µm was achieved at an 8-bit grayscale resolution in the



reconstructed tomogram. The energy settings of the subsamples were 125 kV and 95 µA without a filter and a spatial resolution of 10 µm was achieved.

Image processing and analysis were done with the open source software packages FIJI ImageJ V1.53 (Schindelin et al., 2012) and QuantIm (Vogel et al., 2010). The protocol mainly followed the procedure described in detail by Schlüter et al. (2016).

First, the 250 cm³ samples were filtered with a 2D non-local means filter with a sigma of 15, as an estimate of the standard deviation of the grey value noise (Buades et al., 2011). Second, vertical differences in average image intensity due to uneven beam intensity were removed. Third, images were segmented into two classes, pore system and soil matrix, using the fuzzy c-means thresholding method (Jawahar et al., 1997;Schlüter et al., 2014).

Total visible macro-porosity, defined as the CT derived porosity, and its vertical variability were determined as the ratio

between pore voxels to the total volume of sample cylinders. Therefore, the same region of interest (ROI 1) with 1600 voxel in diameter and 1000 voxel in height was adjusted to the lower boundary of each sample tomogram (Fig. 1 (c)). The pore size distribution (PSD) was determined for a second ROI (ROI 2, Fig. 1 (c)) placed in the centre of the sample to avoid the contribution by an empty headspace due to the settling of repacked samples by FTC. The PSD was computed by the maximum inscribed sphere method using the LocalThickness method (Fig. 1 (d)) in FIJI ImageJ. The connectivity of the pore network

was described by the connection probability ($\Gamma$) and the Euler number ($\chi$). The $\Gamma$-indicator reflects the probability of two randomly chosen pore voxels to belong to the same pore cluster (Renard and Allard, 2013) and was calculated for the entire sample (ROI 1) by:

$$\Gamma(p) = \frac{1}{n_p^2} \sum_{i=1}^{N(X_p)} n_i^2, \tag{1}$$

where $n_p$ is the total number of pore voxels in the analysed volume $X_p$ and $n_i$ is the number of pore voxels per cluster. $\chi$, which quantifies the connectivity as the number of isolated pores minus the number of redundant connections plus the number of

cavities (Vogel et al., 2010), was calculated with the MorphoLibJ plugin (Legland et al., 2016) and expressed as a density for ROI 2. Structural changes of the solid phase, i.e. the fragmentation of soil clods or the creation of new pores (Fig. 1 (d)), were measured by the Euclidean distance of every voxel classified as soil matrix to the nearest pore (ROI 2). The mean pore size and the mean Euclidean distance of each sample was derived from the respective frequency distributions.

The settings of the subsamples differed slightly to those of the larger volumes. Here, a sigma of 10 was used to remove image

noise and correction of vertical image intensity was not necessary. The contrast between matrix and pores was enhanced by an unsharp mask (Schlüter et al., 2014). Image segmentation based on the maximum variance between image classes (Otsu, 1979). The various structural parameters for the entire subsample volume were determined as described above.

## 2.3 Soil hydraulic properties

The soil water retention curve and the unsaturated hydraulic conductivity were measured using the HYPROP device (Meter

Group, USA) based on the evaporation method (Schindler et al., 2010). For each soil texture, five undisturbed FTC samples and three undisturbed controls were saturated and analysed for a pF range from 0 to 3.7. For the repacked samples (five R, and



five R control), a saturation of the sample was not possible due to their fragile soil structure. Here, the measurement of hydraulic properties started with the initial water content which was adjusted to 10 hPa (pF 1). The water content and hydraulic properties at pF 1.8, pF 2.5, and pF 3.7 were used for statistical analysis. At the end of the experiments, the soil cores were dried at 105°C
to determine the bulk density and the total porosity $\phi$.

## 2.4 Statistical analysis

Data management, data analysis and figures were done using the open source packages tidyverse (Wickham et al., 2019) and ggplot2 (Wickham, 2016) in R Version 4.0.2 (R Core Team, 2020). The comparisons of means for each soil property as a function of FTC were done using a one-way ANOVA for repeated measures in the rstatix package (Kassambara, 2020).
Parameters were tested for normality and homogeneity of variance for each time point by the Shapiro-Wilk test and Levene's test, respectively. The p-values of the pairwise t-tests were adjusted using the Bonferroni multiple testing correction method. Mean soil parameters determined at the end of the experiment, hydraulic properties and structural characteristics of the subsamples, were checked for equal variance (F-test) and compared by Welch two sample t-test (control vs. FTC 19). The model fitting for trend analysis of the frequency distribution, i.e. the pore size distribution and pore distance, was done with
the ggeom_smooth function in ggplot2, a local fitting of a polynomial surface determined by one or more numerical predictors. $\Gamma$ is a probability distribution, hence a statistical analysis with the used test models was not possible.

## 3 Results

### 3.1 Soil hydraulic properties

The bulk density and thus the total porosity $\phi$ changed significantly after 19 FTCs for the repacked silt clay from 0.70 g cm$^{-3}$
($\phi$ = 73.7 vol.-%) to 0.75 g cm$^{-3}$ ($\phi$ = 71.8 vol.-%), while the other treatments were not significantly affected (Table 1). In the repacked samples 55 % (silt clay) to 35 % (silty loam) of the total pore volume was already drained at -10 hPa, the starting condition of the measurement. Thus, potential differences in water retention close to saturation were not detected. Figure 2 shows the soil water retention curve (a) and the unsaturated hydraulic conductivity (b) as a function of the negative logarithm of matrix potential [pF]. With the exception of one repacked silty loam sample, a sample which was compacted most due to
FTCs, the water retention curve shows little variability within the treatments. After 19 FTCs (Fig. 2, orange), freezing and thawing significantly changed hydraulic properties of both soils textures and initial structures (Table 1). Compared to the controls (Fig, 2, green), water retention of the repacked silt clay was significantly reduced at pF 2.5 and for the undisturbed silt clay at pF 3.4. No significant differences in water retention were observed for the silty loam. In contrast, the unsaturated hydraulic conductivity of all treatments was significantly increased over a wide range from pF 2.0 to pF 3.0. The impact of 19
FTCs on soil hydraulic properties was significantly more pronounced for the unsaturated conductivity than for the water retention curve, especially for the silt clay.




**Table 1: The mean total porosity (ϕ) and the mean soil hydraulic properties at different potentials, i.e. the water retention (θ) and unsaturated hydraulic conductivity (kᵤ). The level of statistical significance difference between FTC treatment and control within a soil texture and treatment determined by a t-test refers to * = p≤0.05, ** = p<0.01, and *** = p<0.001.**

| | silt clay | | | | silty loam | | | |
|---|---|---|---|---|---|---|---|---|
| | repacked | | undisturbed | | repacked | | undisturbed | |
| | control | FTC 19 | control | FTC 19 | control | FTC 19 | control | FTC 19 |
| | (n=5) | (n=5) | (n=3) | (n=5) | (n=5) | (n=5) | (n=3) | (n=5) |
| $\phi$ [vol.-%] | 73.7 | 71.8** | 60.1 | 60.2 | 68.1 | 67.8 | 44.9 | 45.0 |
| $\theta$ at pF 1.8 [vol.-%] | 29.4 | 29.5 | 46.1 | 47.2 | 25.8 | 27.4 | 33.7 | 34.3 |
| $\theta$ at pF 2.5 [vol.-%] | 25.2 | 23.4*** | 39.1 | 36.8 | 22.2 | 22.7 | 28.5 | 28.9 |
| $\theta$ at pF 3.7 [vol.-%] | 19.0 | 17.7 | 31.3 | 26.6** | 14.8 | 14.5 | 16.1 | 16.0 |
| $K_u$ at pF 2.5 log[cm/d] | -2.99 | -2.47*** | -3.37 | -2.96*** | -1.90 | -1.44** | -1.95 | -1.50** |
| $K_u$ at pF 3.4 or 3.8 log[cm/d] | -4.74 | -4.75 | -4.65 | -4.82* | -4.48 | -4.46 | -4.51 | -4.55 |


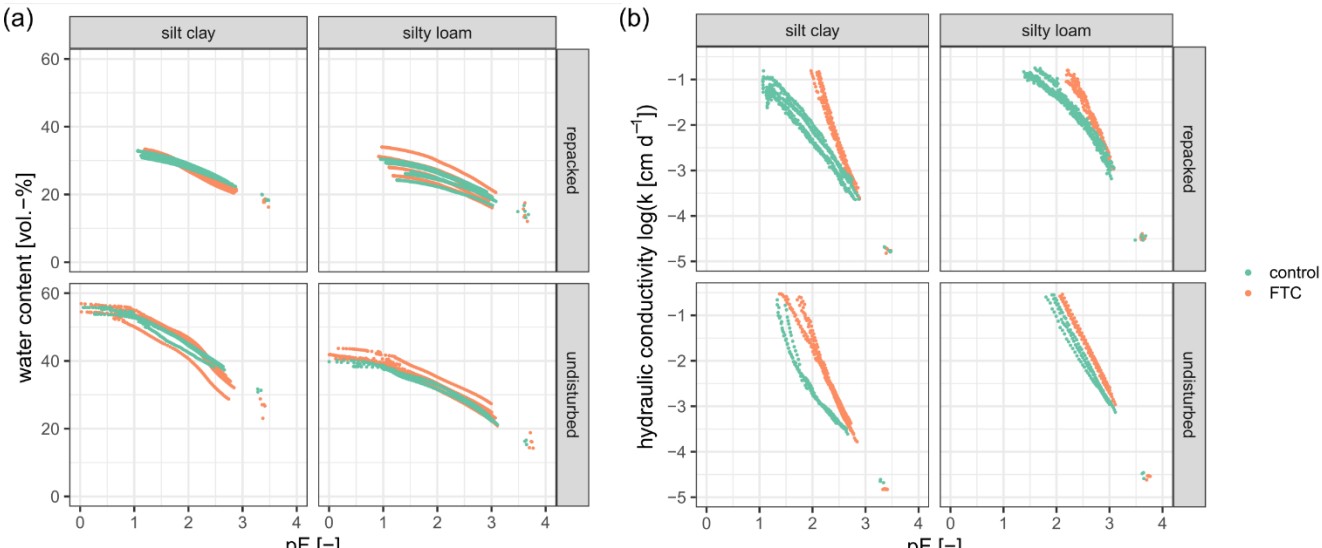

**Figure 2: Soil water retention curve (a) and unsaturated hydraulic conductivity (b) determined for the control samples (green) and after 19 FTCs (orange) for two different soil structures: repacked (top), undisturbed (bottom), and two different soil textures: silt clay (left) and silty loam (right).**

**3.2 Soil structure development throughout the experiment**

Multiple FTC significantly changed the mean soil structure characteristics (≥ 48 µm) of the repacked samples (Table 2), while in the undisturbed samples the changes occurred mainly in certain areas or specific size ranges (Fig. 3). The profiles of visible porosity from column top to bottom (Fig. 3 (a)) and the decrease in mean porosity at the centre of the samples (Table 2) indicate that the repacked samples of both soil textures underwent settling and compaction. The measured visible porosity (silt clay

29.8 vol.-%, silty loam 26.9 vol.-%) decreased sharply at the beginning of the experiment (silt clay -4.7 vol.-% after 2 FTCs, silty loam -1.26 vol.-% after 5 FTCs), while compaction in subsequent FTCs was less pronounced. For the silty clay, another





significant decrease was determined after 10 FTCs (-8.1 vol.-%) and after 19 FTCs (-10.2 vol.-%). In contrast, the mean determined porosity of the repacked silty loam was not significantly changing after 5 FTCs. Settling and compaction was not observed for the undisturbed samples. Here, visible porosity slightly increased at the sample top, which can be assigned to the

formation of a platy soil structure determined by visible inspection of the X-ray CT images (supplementary information, S3). This was particularly pronounced for the silty loam and increased with number of FTCs. In the centre of the samples, a significant increase in macro-porosity was determined after 2 and 19 FTCs for the silty loam, while the silt clay was not affected.

**Figure 3: Soil structure characteristics determined at a resolution of 48 µm as after multiple freeze-thaw cycles (FTC): the mean visible porosity profile over the sample height (a), the Euler-Number χ (b), the mean pore size distribution and their fits (c), and the**





**mean pore distance distribution and their fits (d). The shaded area in (a) and the point ranges in (c) and (d) mark the 95 % confidence interval of the represented data (2* standard error), the latters in (b) depict significant differences within a treatment (n. s. = not significant). Note that the y achses in (b) differ for repacked and undisturbed samples.**

Settlement and compaction had different effects on the connectivity parameters of the treatments. The mean initial Γ-Indicators, which indicates the probability for a connected pore system, were close to 1 for the repacked samples of both textures, 0.86 for the undisturbed silt clay, and 0.79 for the undisturbed silty loam, mainly reflecting differences in initial macro-porosity (Table 2). These Γ values close to 1 indicate that all treatments had a well-connected pore system. During the experiment, the Γ-Indicator was slightly reduced for the repacked samples. The Euler-Number χ increased systematically with FTC for the

repacked samples and varied considerably for the undisturbed samples (Fig. 3 (b), Table 2). χ is more sensitive to subtle changes in pore connectivity, as it does not only reflect if a pore structure is well or poorly connected, but by how many connection or into how many isolated pores. An increase of the Euler number can thus be associated with an increase of isolated pores, an Euler Number below 0 describes a well-connected pore system, above 0 a poorly connected pore system. With the exception of the repacked silt clay, the pore systems ≥ 48 μm of all other treatments were poorly connected. The connectivity

of the repacked silt clay decreased continuously with the number of FTCs. The connectivity of the repacked silty loam decreased significantly after 2 FTCs and after 10 FTCs. For the undisturbed silt clay, no significant effect of FTCs on χ was observed, while for the silty loam, the connectivity was first slightly decreasing (5 FTCs) and then increasing (10 and 19 FTCs). Both treatments showed a high variability among samples compared to the control.

  The decrease in porosity and connectivity was accompanied by a reduction in mean pores sizes in the centre of the samples. A

significant reduction from 0.99 mm to 0.64 mm was observed for the repacked silt clay throughout the experiment. For the silty loam, the mean pore size was decreasing from 1.48 mm to 1.30 mm, mainly after 2 FTCs. For the undisturbed samples, the reduction from 0.68 mm to 0.60 mm for the silt clay and from 1.07 mm to 0.90 mm for the silty loam was gradual and not significant. The pore size distribution in Fig. 3 (c) shows that the decrease in pore size was caused by a shift from larger macro-pores to smaller macro-pores. The frequency of pores larger than 0.5 mm and 1 mm decreased systematically with increasing

FTC number for the repacked silt clay and silty loam, respectively. At the same time the frequency of smaller macro-pores increased continuously. A slightly increasing frequency of smaller macro-pores less than 0.5 mm in diameter was also observed for both undisturbed treatments.

  The fragmentation of soil clods and macro-aggregates was measured by the average distance of all soil voxels to the nearest pore. For all treatments, the mean pore distance was found to decrease with increasing number of FTCs. For the silty loam, a

significant decrease with reference to the initial structure was observed after 10 FTCs for the undisturbed, and after 2 and 10 FTCs for the repacked samples. The initial variability for the repacked silt clay was high, hence a significant reduction was determined between 2 to 5 FTCs. The frequency distribution of pore distances in Fig. 3 (d) shows that for the repacked samples there was a significant decrease in frequency of compact soil areas with pore distances larger than 0.6 to 0.8 mm. For the undisturbed samples, compact soil areas with distances larger than 0.4 mm were fragmented.





**Table 2: Soil structural parameters as arithmetic means and their standard errors determined via X-ray µCT at a resolution of 48 µm. The Gamma-Indicator describes the connectivity for the entire sample. Γ was not statistically tested. The porosity, Euler number χ, mean pore size, and mean pore distance were determined for the centre of the samples (Figure 1 (c), ROI 2). The latters depict significant differences within a treatment.**

### silt clay

| | repacked FTC (n=7) | | | | | repacked control (n=5) | | | | | undisturbed FTC (n=5) | | | | | undisturbed control (n=3) | | | | |
|---|---|---|---|---|---|---|---|---|---|---|---|---|---|---|---|---|---|---|---|---|
| | FTC 0 | FTC 02 | FTC 05 | FTC 10 | FTC 19 | FTC 0 | FTC 02 | FTC 05 | FTC 10 | FTC 19 | FTC 0 | FTC 02 | FTC 05 | FTC 10 | FTC 19 | FTC 0 | FTC 02 | FTC 05 | FTC 10 | FTC 19 |
| Porosity ≥48 µm [vol.-%] | 29.79[a] (±2.0) | 25.09[b] (±1.9) | 23.28[b] (±1.6) | 21.74[c] (±1.5) | 19.57[d] (±1.5) | 32.65[a] (±0.9) | 32.29[a] (±0.7) | 32.42[a] (±0.7) | 32.56[a] (±0.7) | 32.04[a] (±0.8) | 7.73[a] (±1.0) | 7.90[a] (±1.0) | 8.11[a] (±1.1) | 7.88[a] (±1.0) | 7.65[a] (±1.1) | 6.95[a] (±0.4) | 6.96[a] (±0.3) | 7.10[a] (±0.4) | 7.01[a] (±0.3) | 7.02[a] (±0.3) |
| Γ-Indicator [-] | 0.99 (±0.0) | 0.99 (±0.0) | 0.99 (±0.0) | 0.99 (±0.0) | 0.98 (±0.0) | 0.99 (±0.0) | 0.99 (±0.0) | 0.99 (±0.0) | 0.99 (±0.0) | 0.99 (±0.0) | 0.86 (±0.0) | 0.88 (±0.0) | 0.88 (±0.0) | 0.88 (±0.0) | 0.83 (±0.0) | 0.85 (±0.0) | 0.84 (±0.0) | 0.85 (±0.0) | 0.85 (±0.0) | 0.85 (±0.0) |
| χ [$mm^{-3}$] | -0.19[a] (±0.0) | -0.17[a] (±0.0) | -0.09[a] (±0.0) | -0.01[bc] (±0.0) | 0.16[c] (±0.1) | -0.25[a] (±0.0) | -0.25[a] (±0.0) | -0.24[a] (±0.0) | -0.25[a] (±0.0) | -0.25[a] (±0.0) | 0.34[a] (±0.1) | 0.30[a] (±0.1) | 0.35[a] (±0.1) | 0.37[a] (±0.1) | 0.58[a] (±0.1) | 0.55[a] (±0.1) | 0.54[a] (±0.1) | 0.57[a] (±0.1) | 0.57[a] (±0.1) | 0.57[a] (±0.1) |
| Mean pore size [mm] | 0.99[a] (±0.0) | 0.84[b] (±0.0) | 0.77[c] (±0.0) | 0.72[c] (±0.0) | 0.64[d] (±0.0) | 0.91[a] (±0.0) | 0.90[a] (±0.0) | 0.90[ab] (±0.0) | 0.90[ab] (±0.0) | 0.88[b] (±0.0) | 0.68[ab] (±0.1) | 0.67[a] (±0.1) | 0.65[ab] (±0.1) | 0.62[b] (±0.1) | 0.60[b] (±0.1) | 0.56[a] (±0.1) | 0.57[a] (±0.1) | 0.56[a] (±0.1) | 0.55[a] (±0.1) | 0.55[a] (±0.1) |
| Mean pore distance [mm] | 0.69[ab] (±0.1) | 0.57[a] (±0.1) | 0.46[b] (±0.1) | 0.44[ab] (±0.1) | 0.37[ab] (±0.0) | 0.49[a] (±0.0) | 0.51[a] (±0.0) | 0.53[a] (±0.1) | 0.54[a] (±0.1) | 0.54[a] (±0.1) | 0.51[a] (±0.0) | 0.52[a] (±0.0) | 0.48[a] (±0.0) | 0.49[a] (±0.0) | 0.45[a] (±0.0) | 0.46[ab] (±0.1) | 0.46[a] (±0.1) | 0.44[ab] (±0.1) | 0.45[ab] (±0.1) | 0.44[b] (±0.1) |

### silty loam

| | repacked FTC (n=7) | | | | | repacked control (n=5) | | | | | undisturbed FTC (n=5) | | | | | undisturbed control (n=3) | | | | |
|---|---|---|---|---|---|---|---|---|---|---|---|---|---|---|---|---|---|---|---|---|
| | FTC 0 | FTC 02 | FTC 05 | FTC 10 | FTC 19 | FTC 0 | FTC 02 | FTC 05 | FTC 10 | FTC 19 | FTC 0 | FTC 02 | FTC 05 | FTC 10 | FTC 19 | FTC 0 | FTC 02 | FTC 05 | FTC 10 | FTC 19 |
| Porosity ≥48 µm [vol.-%] | 26.94[a] (±1.3) | 26.01[ab] (±1.2) | 25.68[a] (±1.2) | 26.06[ab] (±1.2) | 25.95[ab] (±1.1) | 29.03[a] (±2.8) | 28.68[a] (±2.8) | 28.63[a] (±2.6) | 29.09[a] (±2.5) | 28.70[a] (±2.6) | 7.48[a] (±0.8) | 8.24[b] (±0.8) | 8.13[abc] (±0.9) | 8.39[abc] (±0.8) | 8.59[c] (±0.9) | 7.06[a] (±2.8) | 7.25[a] (±2.8) | 6.98[a] (±2.7) | 7.32[a] (±2.7) | 7.35[a] (±2.6) |
| Γ-Indicator [-] | 0.99 (±0.0) | 0.98 (±0.0) | 0.98 (±0.0) | 0.98 (±0.0) | 0.97 (±0.0) | 0.99 (±0.0) | 0.99 (±0.0) | 0.99 (±0.0) | 0.99 (±0.0) | 0.99 (±0.0) | 0.79 (±0.0) | 0.82 (±0.0) | 0.81 (±0.0) | 0.79 (±0.0) | 0.76 (±0.0) | 0.78 (±0.0) | 0.79 (±0.0) | 0.80 (±0.0) | 0.81 (±0.0) | 0.81 (±0.0) |
| χ [$mm^{-3}$] | 0.10[a] (±0.0) | 0.38[bc] (±0.0) | 0.32[b] (±0.0) | 0.50[c] (±0.0) | 0.59[c] (±0.0) | 0.09[a] (±0.0) | 0.06[a] (±0.0) | 0.04[a] (±0.0) | 0.05[a] (±0.0) | 0.03[a] (±0.0) | 0.79[ab] (±0.1) | 0.83[ab] (±0.1) | 0.68[a] (±0.1) | 0.93[b] (±0.1) | 1.00[b] (±0.1) | 0.64[a] (±0.0) | 0.62[a] (±0.0) | 0.65[a] (±0.0) | 0.72[a] (±0.1) | 0.78[a] (±0.1) |
| Mean pore size [mm] | 1.48[a] (±0.1) | 1.38[b] (±0.1) | 1.36[b] (±0.1) | 1.33[b] (±0.2) | 1.30[b] (±0.2) | 1.22[a] (±0.0) | 1.22[a] (±0.0) | 1.22[a] (±0.0) | 1.22[a] (±0.0) | 1.21[a] (±0.0) | 1.07[a] (±0.3) | 1.01[a] (±0.3) | 1.01[a] (±0.3) | 0.95[a] (±0.3) | 0.90[a] (±0.2) | 1.49[a] (±0.3) | 1.56[a] (±0.3) | 1.46[a] (±0.3) | 1.47[a] (±0.3) | 1.49[a] (±0.4) |
| Mean pore distance [mm] | 0.61[a] (±0.0) | 0.37[b] (±0.0) | 0.38[b] (±0.0) | 0.34[c] (±0.0) | 0.32[c] (±0.0) | 0.45[a] (±0.0) | 0.49[a] (±0.1) | 0.55[a] (±0.1) | 0.56[a] (±0.1) | 0.60[a] (±0.1) | 0.45[a] (±0.0) | 0.39[ab] (±0.0) | 0.41[a] (±0.0) | 0.36[b] (±0.0) | 0.33[b] (±0.0) | 0.53[a] (±0.0) | 0.55[a] (±0.0) | 0.55[a] (±0.0) | 0.52[a] (±0.0) | 0.50[a] (±0.0) |





### 3.2 Soil structure of subsamples after 19 FTCs

The main effects on soil structure determined at a resolution of 48 µm occurred at the lower resolution boundary of the X-ray µCT scans. However, structural features smaller than 4 to 5 voxels are systematically underestimated due to image processing (Vogel et al., 2010;Leuther et al., 2019). Hence, subsamples taken at the end of the experiment extended the information of soil structure characteristics at the lower resolution limit of the larger experimental containers. A resolution of 10 µm further describes the pore system which determine hydraulic properties at around pF 2, the pF value at which all FTC-treatments had

an increased hydraulic conductivity.

Statistical significant differences in pore structure between the subsamples from the control and the samples after 19 FTCs were found mainly in the connectivity parameters (Table 3). With the exception of the repacked silty loam, the Euler-Number χ was significantly reduced in all treatments after 19 FTC, implying an increase in redundant connections in the pore network. The absolute χ values increased by a factor of 2 to 3 compared to 250 cm³ samples, as the chance of detecting small isolated

pores increases with resolution. For the undisturbed silty loam, Γ was increasing from 0.82 (control) to 0.89 (FTC) as a result of a significant increase in macro-porosity ≥ 10 µm (13.9 to 16.4 vol.-%, respectively). The connection probabilities of all other treatments ranged from 0.90 for the undisturbed control of the silt clay to 0.98 for the repacked silt clay (FTC), corresponding to a very well connected pore system. No additional increase in porosity were found for these treatments.

**Table 3: Soil structural parameters as arithmetic means and their standard errors determined via X-ray µCT at a resolution of 10**
**µm. The level of statistical significance difference between FTC treatment and control within a soil texture and treatment determined by a t-test refers to * = p≤0.05, ** = p<0.01, and *** = p<0.001. Γ was not statistically tested.**

| | silt clay | | | | silty loam | | | |
|---|---|---|---|---|---|---|---|---|
| | repacked | | undisturbed | | repacked | | undisturbed | |
| | control (n=6) | FTC 19 (n=7) | control (n=7) | FTC 19 (n=7) | control (n=8) | FTC 19 (n=8) | control (n=8) | FTC 19 (n=10) |
| Porosity ≥10 µm [vol.-%] | 32.49 (±4.11) | 28.74 (±2.52) | 14.22 (±1.09) | 14.00 (±0.98) | 33.26 (±2.37) | 36.05 (±4.39) | 13.93 (±0.62) | 16.40* (±0.74) |
| Γ-Indicator [-] | 0.97 (±0.01) | 0.98 (±0.00) | 0.90 (±0.01) | 0.91 (±0.01) | 0.93 (±0.01) | 0.94 (±0.03) | 0.82 (±0.01) | 0.89 (±0.01) |
| χ [mm$^{-3}$] | 18.05 (±2.96) | -2.80*** (±2.54) | 66.38 (±8.47) | 45.34* (±3.54) | 142.39 (±10.24) | 140.94 (±11.66) | 274.30 (±23.53) | 204.36* (±15.27) |
| Mean pore size [mm] | 0.41 (±0.06) | 0.40 (±0.03) | 0.29 (±0.06) | 0.27 (±0.05) | 0.76 (±0.08) | 0.69 (±0.06) | 0.18 (±0.01) | 0.20 (±0.01) |
| Mean pore distance [mm] | 0.08 (±0.00) | 0.08 (±0.00) | 0.09 (±0.00) | 0.09 (±0.00) | 0.07 (±0.00) | 0.06 (±0.00) | 0.05 (±0.00) | 0.06 (±0.00) |

19 FTCs induced an increase in the volume fraction of pore diameters of < 0.1 mm (Fig. 4 a). However, the effect was too small to significantly reduce the mean pore size in any of the treatments, partly due to high variability among the replicates

285    (Table 3). This was also the case for the mean pore distance, where the volume fraction of distances < 0.05 mm was slightly





increased for most of the FTC treatments, with the exception of the undisturbed silty loam (Fig. 4 (b)). However, these small effects on the pore distance distributions were not reflected by the mean pore distances. Here, no significant effects were determined for the subsamples taken after 19 FTCs.

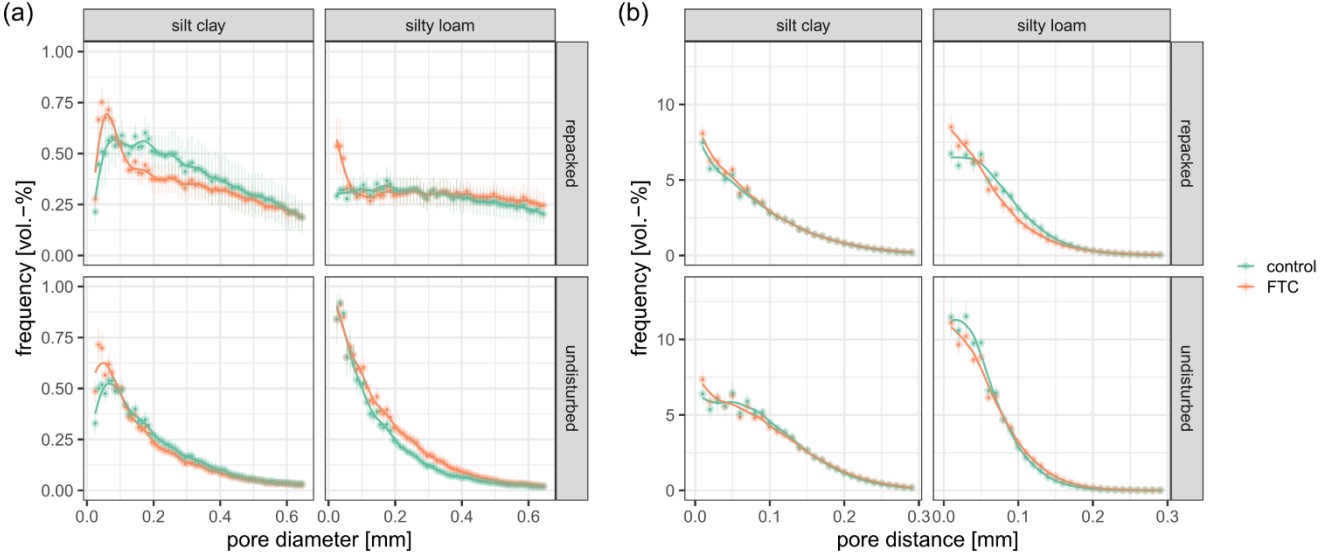

Figure 4: The mean pore size distribution (a) and the mean pore distance distribution (b) determined at a resolution of 10 μm. The point ranges mark the 95 % confidence interval of the represented data (2* standard error). Data from the control are shown in green, data obtained from subsamples taken after 19 FTCs are shown in orange.

**4 Discussion**

The combination of X-ray μCT and hydraulic property measurements allowed the detection of soil structural changes over a wide range of pore diameters. Multiple freezing and thawing of soils changed soil structural and soil hydraulic properties of all investigated treatments. A very loose structure at the direct soil surface resembled by repacked soil clods taken from a ploughed field was more sensitive to FTC than an intact soil structure under grassland which has developed over decades. This is consistent with field measurements of soil stability under different management systems, where ploughed soils layers showed lower resistance to physical stresses compared to reduced or no-till treatments (Schjønning and Rasmussen, 1989;Wiermann et al., 2000;Munkholm et al., 2008). Regarding the impact of FTCs on soil structure, Ma et al. (2019) concluded that well-structured soils are more difficult to be fragmented by FTC than degraded or compacted soils. In our experiments, settling and compaction of soil clods were the main factors for changes in the macro-pore system of the repacked samples, with silt clay (40 % of clay content) being more affected than silty loam (21 % of clay content). Multiple FTCs caused a significant reduction in porosity, mean pore size, and connectivity of the macro-pore system, which is schematically shown in Fig. 5 (a) and (b). The reduction occurred gradually with increasing number of FTCs, so that already 2 to 5 FTCs had a significant influence on the investigated structural parameters. Such a low number of FTCs occurs under natural conditions



directly at the soil surface at both sites (2017: 13 and 15 FTCs; 2018: 7 and 16 FTCs in 5 cm depth, Fig. S1 and S2) but is barely reached in deeper layers (1 to 3 FTCs in 10 and 20 cm depth).

In addition to the collapse of the fragile macro-structure, new pores formed at the transition from small macro-pores to meso-
pores (pF 2-3). On the one hand, this was caused by the compaction of the samples, where a continuous shift of larger macro-pores towards smaller macro-pores was observed (Fig. 5). On the other hand, new pores were developed by the formation of cracks that fragmented larger soil clods. This is consistent with Ma et al. (2019), where the amount of small aggregates increased after multiple FTC (up to 30) compared to the control. Surprisingly the fragmentation of soil as indicated by the reduction of pore distances, was more pronounced for the silty loam than for the silt clay with larger swell-shrink dynamics.
Lehrsch (1998) found an increase in aggregate stability after 1 or 2 FTCs, but it was also shown that macro-aggregates with high clay contents were less vulnerable to FTCs than a loam and silt loam. Dagesse (2013) also found, that clay content is an important factor determining the extent of freeze-induced desiccation process (improvement of aggregate stability) and thaw-induced degradation of aggregate stability due to attendant liquid water.

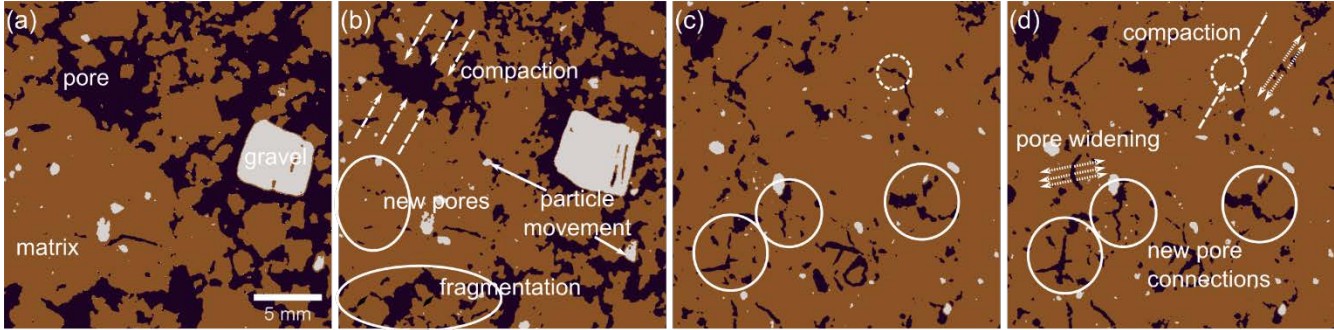

**Figure 5: Scheme of soil structure transformation of a repacked sample after 0 (a) and 19 FTCs (b), and of an undisturbed sample after 0 (c) and 19 FTCs (d).**

For the intact, well-structured soils samples taken under grassland, changes in pore size distribution and pore distance were observed mainly in the silty loam. Here, a platy soil structure with horizontal orientation developed in the uppermost 5 to 10 mm of the columns (Fig. S3). Taina et al. (2013) observed these horizontally elongated aggregates and planar voids in the
topsoil after freezing and thawing across different textures and farming practices. They are typical for low cooling rates when water is moving towards the ice front and surrounding soil areas are desiccated. This promotes the formation of new ice-nucleation centres in wet soil below, and consequently elongated soil aggregates between ice lenses which are regarded to be stable structural elements (Taina et al., 2013;Van Vliet-Lanoë and Fox, 2018). When analysing the soil structure in the centre of the undisturbed samples, only small effects on the pore-size distribution, porosity, and connectivity of the macro-pore
network by FTCs were observed, but a significant fragmentation of soil fragments larger 0.5 mm after 10 FTCs. The results indicate that two different processes are taking place at the same time, which are opposing each other and thus keep the determined pore parameters rather constant (Fig. 5 (c) and (d)). On the one hand, the fragmentation of denser soil areas produced new pores and connections which were at the lower resolution limit of this study (48 µm). On the other hand it also

caused a compaction and reduction of the already existing pore volumes and disconnect existing macro-pores. For aggregates,
Six et al. (2004) have described these opposing processes of degradation and stabilisation as well, where stability measurements
of bulk samples yielded in an average of both processes.

Starkloff et al. (2017) analysed the impact of 1 to 6 FTCs on a pore system larger than 140 µm with X-ray µCT. The undisturbed
soil cores taken from a ploughed layer experienced 6 FTCs and much higher and faster temperature gradients between freezing
(-15°C) and thawing (40°C) due to scan time limitations. For a loamy sand, slight settlement of samples was observed,
accompanied by a decrease in macro-porosity for the pore sizes investigated ($\geq$ 0.3 mm). For a silty clay loam, no settlement
was observed but a shift from larger macro-pores ($\geq$ 1.9 mm) towards smaller macro-pores (0.4 - 1.5 mm). By visible
inspection, the authors observed the development of small cracks and voids which were below the detection limit of their
image analysis method. No significant effects on pore connectivity were determined. The presented results in our study, based
on a higher resolution, confirm the trends on soil structure development identified by Starkloff et al. (2017) which were limited
in resolution and in number of FTCs at the time. Also in the current experiments, many of the described changes in soil
structure were detected at the lower resolution boundary of the X-ray µCT scans and only revealed by subsampling at the end
of the experiment. The information obtained at two different resolutions shows that FTC confirm structural changes towards
a well-connected pore system for pores smaller 0.5 to 0.8 mm in diameter.

This was corroborated by the increase in unsaturated hydraulic conductivity between pF 2 to pF 3 for all samples. In
comparison, water retention in the same range of matrix potential was only slightly affected for the repacked samples. This
indicates that the pore size distribution from 0.002 to 0.05 mm (according to Young-Laplace-equation) was only slightly
affected, but that the connectivity of these pores increased significantly after 19 FTCs. Most studies investigated the impact of
FTC on saturated hydraulic conductivity and measured its increase which was mainly related to the creation of vertical voids
(Qi et al., 2006). After 30 FTCs, Ma et al. (2019) found no consistent effects of FTC on soil water retention curves among
different soils but in total a significant increase in plant available water content for all tested treatments. They related the
increase to changes in the pore size distribution. Since the determined pore structure characteristics still changed after 19 FTCs,
there is a possibility that another 11 FTCs could have had a greater impact on the water retention of our tested soils.

**4 Conclusion**

Freezing and thawing of soils has a large impact on soil structure development in areas where air temperature is fluctuating
around the freezing point. It was shown that multiple FTC fragmented large soil clods and intact soil. Hence, it is an important
factor for farmers to modify the soil surface for seedbed preparation and potentially to loosen compacted soil areas or
consolidate very loose soil. The creation of a well-connected pore system in the diameter range of meso-pores to narrow macro-
pores increased the unsaturated hydraulic conductivity, independent of the soil texture and the initial soil structure. This is a
beneficial feature for the soil quality in spring, as the water transport of melting snow is increased and the seedbed has an
improved hydraulic connection to the soil underneath.





In prospect of milder winters due to climate change, soils in the mid-latitudes may experience a decreasing number of FTCs in winter. This could lead to farmers having to spend more time on field preparation in spring and to a degradation of soil structure over years. We could show that already 2 to 5 FTCs significantly affected the soil structure and that the effects per FTC on the tested parameters was decreasing with the number of FTC. However, it was also shown that changes were still

accumulating after multiple FTCs, and that further significant changes occurred after 10 and 19 FTCs. While the impact on the macro-pore network was largest for the silt clay, the loamy sand was more susceptible to the fragmentation of soil compartments. Even though, the top soil of intact grassland can be considered to have reached a soil structure in dynamic equilibrium that has experienced many FTCs in the making, soil structure is still affected by FTC.

**Data availability**

The data used in this study are available from the corresponding author upon request.

**Author contributions**

FL designed and carried out the experiments with contributions from SS. Both authors developed and evaluated the X-ray μCT

image analysis steps. FL prepared the manuscript with contributions from SS.

**Competing interests**

The authors declare that they have no conflict of interest.

**Acknowledgments**

This work was supported by the German Research Foundation (DFG), grant number 416883305. We would like to thank Kristina Kleineidam from the University of Giessen for giving us the opportunity to take soil samples at the long-term study site in Giessen. A special thanks to Max Köhne for his help and support in lab work, especially during Covid-19 pandemic.

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
