# Peer review of "Impact of freeze-thaw cycles on soil structure and soil hydraulic properties"

_SOIL, 2021_

## Author Response (AR1)

**RC1:**

The manuscript experimentally investigated the effect of freeze-thaw cycles on the pore structure and hydraulic conductivity of two different soils. The manuscript is well-written, and easy to follow. The experimental materials and setup are clearly described, and the testing results are reasonably analyzed and discussed.
A minor revision is suggested. Several comments are given below.

*Answer: We thank the reviewer for the positive feedback, comments and suggestions. Given lines in the answers refer to the revised manuscript without tracked changes.*

Line 6: The abstract is relatively long, and some minor details may be deleted to make it concise.

*Answer: We have shortened the abstract by deleting some minor details.*

Line 119: Only one thermometer was inserted into one sample, and the measured temperature value was used as a representative for all the samples. Is this reasonable enough?

*Answer: Yes, there was only one dummy sample with a thermometer per experiment. However, the large number of replicates required that the experiment was carried out in two runs and the temperature curve in the second run measured within a new repacked sample was comparable. We have chosen to show only one temperature curve, as the figure is more comprehensible for the reader this way. The condition (frozen or thawed) of the freezer packs at the end of the cycle also served as a control for thawing. We have now included the information, that the shown time serious is one* representative *example (Line 123). To clarify the experimental setup, we have rephrased the section about the temperature control (Lines 120 -127).*

Line 122&Figure 1(b): When the measured soil temperature was just above 0.5 deg C, frozen part may still exist inside the soil sample (which means the sample was not fully thawed). As a result, this may not represent a full freeze-thaw cycle.

*Answer: In preliminary tests, we investigated the status of the sample at different measured temperatures. It was found that the samples and freezer packs were thawed at 0.5°C (measured inside the sample) and that the sample was completely frozen when temperature was below -2.0°C. Therefore, these temperatures were our thresholds for one cycle. In addition, the sensor of the thermometer integrates the temperature over a depth of 1.0 cm (20 % of the maximum sample height) and was installed in the centre of the sample. So it should be representative for the locations that freeze and thaw last. In general, the freeze-thaw temperature was selected based on the recommendations of Henry (2007) as described in Lines 85-87.*
*To clarify the experimental setup, we have rephrased the section about the temperature control (Lines 120 -127) and further, marked the temperature thresholds in Figure 1b.*

Line 128-129: The subsamples were air dried at room temperature. During this process, the soil structure and pore structure should have changed. Did the authors distinguish this from the soil structure change due to effect of freeze-thaw cycles?

*Answer: To avoid changes in pore and soil structure due to drying and clay mineral shrinkage, the subsamples were only dried until narrow macropores >10 were drained (-300 hPa), not completely air dried. To do so, we calculated the water loss based on the measured water retention curves and carefully observed the water loss throughout the drying process on a sensitive precision balance. We have added an explanation to the new manuscript (Line 134).*

Line 130-131: This sentence is unclear.

*Answer: We have rephrased both sentences to make the drying process of the subsamples more clear (Lines 132 - 134).*

Line 210: Was the structure change uniform along the sample's profile? Was the effect of freeze-thaw cycle monotonic with the increasing FTC number? Or, did the measure soil properties show variations (rather than monotonically decrease/increase) with the increasing number of FTC? What are the reasons?

*Answer:*
*Settling and compaction were the main drivers for changes along the profiles. When comparing the visible pore volume as a function of the sample height in Figure 3 (a), it is shown that the effect was dependent on the soil texture and initial structure. For the repacked silt clay, the first 2 FTC already increased the visible porosity at the upper sample boundary (30-45 mm) drastically, after which the settlement per FTC decreased. For the repacked silt clay, the largest effect per FTC was measured between 2 to 5 FTC, after which the measured differences became smaller but more monotonous (when comparing 10 and 19 FTC). For the undisturbed samples, changes occurred at the lower resolution boundary and settling was prevented by the higher bulk density. Here, a statement on the monotonic behaviour of settlement and compaction was not possible. We have now described the effect of settlement and compaction on the changes along the profile in more detail in the new manuscript (Lines 218 -220).*
*In addition, the effect of all measured structural soil properties as a function of FTC number are presented and discussed in the manuscript. Table 2 and Figure 3 provides the results of a statistic analysis: "The comparisons of means for each soil property as a function of FTC were done using a one-way ANOVA for repeated measures (Line 181)" to determine significant changes after certain numbers of FTC.*
*For example:*
*Line 221: The measured visible porosity decreased sharply at the beginning of the experiment while compaction in subsequent FTCs was less pronounced.*
*Line 311: The reduction occurred gradually with increasing number of FTCs, so that already 2 to 5 FTCs had a significant influence on the investigated structural parameters.*

The size and clearness of the figures may be increased for better presentation.

*Answer: Thank you for the hint. For the final version we will upload the figures in vector format.*

In the discussion part, the detailed description of other researchers' work should be limited. While the analysis and discussion on the authors own findings should be strengthened/highlighted.

*Answer: We tried to implement the different approaches/methods which described the impact of FTC on soil structure of different textures and management practices, which are not consistent. Therefore the proportion of other studies might be slightly larger as usual. In the new manuscript we have slightly shorten the content of other studies and highlighted some of our own results.*

**RC2:**

*Answer: We thank the reviewer for the positive feedback, comments and suggestions. Given lines in the answers refer to the revised manuscript without tracked changes.*

This study was to analyze the cumulative effects of multiple FTC on soil structure and soil hydraulic properties for two different textures and two different initial structures.
I have decided to minor revision for Soil due to the following reasons:

The description of freeze-thaw experiment is not clear enough. It is unrecognizable from Fig. 1B that how long a specific freeze-thaw cycle is. It seems that the temperature in each cycle is not strictly controlled to reach the same temperature.
What is the reason and practical significance author select this freeze-thaw temperature?

*Answer: In preliminary tests, we investigated the status of the sample at different measured temperatures. It was found that the samples and freezer packs were thawed at 0.5°C (measured inside the sample) and that the sample was frozen when temperature was below -2.0°C. Therefore, these temperatures were our thresholds for one cycle. That is, the measured temperatures had to reach at least this values, but were allowed to proceed beyond it. Every third cycle the sample reached a maximum of 4°C, the temperature of the control stored in the refrigerator, due to the weekends. The duration for one cycle was two to three days, as described in the manuscript (Lines 120 -122). In general, the freeze-thaw temperature was selected based on the recommendations of Henry (2007) as described in Lines 83-85.*
*To clarify the experimental setup, we have rephrased the section about the temperature control (Lines 120 -127) and further, marked the temperature thresholds in Figure 1b.*

In this study, the author proposed a concept of pore distance. However, it seems that the pore distance shown in Fig.1d is not consistent with the actual situation.

*Answer: The measurements of pore distances is done for every voxel in all directions in the 3-dimensional image. The calculated pore distance contains also the information based on the segmented pores and the soil matrix of the slices above and below the shown slice of the aggregates. Hence, the pore distance in Fig.1d shows the actual situation in 3D while the segmented slice in 2D can only provide information about the specific height. This was clarified in the revised manuscript (Lines 138-139).*

Line 26: What is the definition of a meso-pore system?

*Answer: We changed the term to "a well-connected pore system of narrow macro-pores" (Line 24), which we measured by X-ray CT and defined by the resolution. In this way, we are in agreement with the official terminology derived from hydraulic properties (micro-pores <0.2µm, meso-pores 0.2-10µm, small macro-pores 10-50µm, macro-pores >50µm). The pore sizes are now specified in the revised manuscript (Lines 81-83).*

Lines 113-116: The effect of freeze-thaw cycles on soil structure would be controlled by soil moisture. The author changed the original soil water content before the experiment, but did not present relevant data.

*Answer: Many studies showed that the water content is one of the controlling factors of soil vulnerability towards freezing and thawing (Lines 55-60). In addition, the potential of water can influence the freezing temperature and the redistribution of water due to its dependence on the hydraulic conductivity. Our intention was to rule out soil moisture a controlling factor and focus on texture and antecedent structure instead. Therefore, a uniform water potential (-10 hPa) was therefore chosen as a reference value. This was also within the water content range of the field-fresh samples taken. Small differences in water content between treatments cannot be avoided by adjusting the water potential in soils with different pF curves. The treatment in the sand bed therefore only led to a small increase in water content of: silt clay undisturbed +0.8 vol.-% (±0.1 vol.-%) / repacked +4.6 vol.-% (±0.4 vol.-%), silty loam undisturbed +6.2 vol.-% (±0.3 vol.-%) / repacked +3.7 vol.-% (±0.8 vol.-%). We have added this information to the revised manuscript (Line 114).*

Figure 3 and 4: I'm confused about the unit of frequency.

*Answer: The image analysis provides the number of voxels which were assigned to the calculated pore sizes/the pore distances. The absolute number depends on image dimensions, which can vary among different samples. In order to make the frequencies comparable, we divide absolute frequencies by number of voxels in the region of interest and show these relative frequencies as volume percentages instead (vol.-%).*

"3.2 Soil structure of subsamples after 19 FTCs", I think it should be "3.3".
Beyond that, the "4 Conclusion" should be "5 Conclusion".

*Answer: Thanks a lot for spotting the mistake. It is corrected in the revised version.*